# Association Between Incident Chronic Kidney Disease and Body Size Phenotypes in Apparently Healthy Adults: An Observational Study Using the Korean National Health and Nutrition Examination Survey (2019–2021)

**DOI:** 10.3390/biomedicines13081886

**Published:** 2025-08-03

**Authors:** Young Sang Lyu, Youngmin Yoon, Jin Hwa Kim, Sang Yong Kim

**Affiliations:** 1Department of Endocrinology and Metabolism, Chosun University Hospital, Chosun University School of Medicine, Gwangju 61453, Republic of Korea; lyu0923@naver.com (Y.S.L.); endocrine@chosun.ac.kr (J.H.K.); 2Division of Nephrology, Department of Medicine, Chosun University Hospital, Chosun University School of Medicine, Gwangju 61453, Republic of Korea; korean8503@chosun.ac.kr

**Keywords:** chronic kidney disease, metabolic syndrome, obese, phenotypes, body composition

## Abstract

**Background/Objectives**: The association between chronic kidney disease (CKD) and body size phenotypes in metabolically diverse but apparently healthy adult populations remains inadequately understood. This study investigated the association between CKD and body size phenotypes in a nationally representative sample of healthy Korean adults. **Methods**: Data from 8227 participants in the 2019–2021 Korean National Health and Nutrition Examination Survey were analyzed. Participants were categorized into four body size phenotypes by combining BMI status (normal weight or obese) with metabolic health status (healthy or abnormal)—MHNW (Metabolically Healthy Normal Weight), MANW (Metabolically Abnormal Normal Weight), MHO (Metabolically Healthy Obese), or MAO (Metabolically Abnormal Obese). CKD was defined based on the urine albumin-to-creatinine ratio and estimated glomerular filtration rate (eGFR). To assess the association between CKD and body size phenotypes, multivariable logistic regression analyses were performed. **Results**: CKD prevalence was 4.4%. MANW and MAO made up 12.6% and 26.4% of the CKD group, compared to 5.0% and 13.2% of the non-CKD group. CKD prevalence by phenotype was observed as follows: MHNW, 3.2%; MANW, 10.5%; MHO, 4.0%; and MAO, 8.5%. CKD odds were highest in the MAO group (OR: 3.770, 95% CI: 2.648–5.367), followed by the MANW (OR: 2.492, 95% CI: 1.547–4.016) and MHO (OR: 1.974, 95% CI: 1.358–2.870) groups. MAO individuals carried a higher CKD risk than MHO individuals (OR: 1.897, 95% CI: 1.221–2.945). **Conclusions**: Among apparently healthy adults, body size phenotypes—particularly those with metabolic abnormalities—were significantly associated with the presence of CKD. These findings highlight the need to assess both metabolic health and body composition for effective CKD prevention and management.

## 1. Introduction

Chronic kidney disease (CKD) has become a major global health concern, contributing significantly to morbidity, mortality, and healthcare costs [1]. Given its asymptomatic nature in early stages and its potential progression to end-stage renal disease or cardiovascular complications, early detection and management of CKD are essential for effective public health strategies [2]. Meanwhile, the definition of obesity is evolving. The 2025 Lancet Commission on Clinical Obesity defines clinical obesity as the presence of excess body fat that leads to adverse health effects [3]. Under this definition, individuals with excessive adiposity may be classified as having clinical obesity if they also exhibit signs of metabolic dysfunction or target organ damage, such as chronic kidney disease (CKD). This framework underscores the importance of evaluating both metabolic health and organ status when determining eligibility for active obesity treatment.

In this context, body size phenotypes—classified by the body mass index (BMI) and metabolic health—have gained prominence as more refined indicators of disease risk [4,5]. The phenotypes are categorized as follows: MHNW, Metabolically Healthy Normal Weight); MANW, Metabolically Abnormal Normal Weight; MHO, Metabolically Healthy Obese; and MAO, Metabolically Abnormal Obese. While obesity has long been linked to CKD, recent evidence suggests that metabolic abnormalities alone, even in normal-weight individuals (MANW), may confer a substantial risk of renal dysfunction [6]. Conversely, the MHO phenotype—historically considered “benign”—may not be metabolically or renally benign if accompanied by subtle organ damage such as CKD [7,8].

Despite growing interest in these phenotypes, population-based evidence examining the association between body size phenotypes and CKD remains limited, especially in Asian populations. Asians differ from Western populations in terms of BMI distribution and metabolic risk profiles, and these differences may affect how phenotypes relate to renal outcomes [9]. South Korea, in particular, represents a timely and relevant setting for such an investigation due to its rapidly aging population and shifting lifestyle patterns that contribute to rising metabolic disease burden.

Therefore, this study aimed to examine the relationship between CKD and various body size phenotypes among apparently healthy Korean adults, utilizing nationally representative data from the 2019–2021 Korean National Health and Nutrition Examination Survey (KNHANES). By incorporating the updated concept of clinical obesity and focusing on metabolic status and organ outcomes rather than body weight alone, this study seeks to provide evidence for improved risk stratification and early intervention strategies. Specifically, the presence of CKD in metabolically abnormal phenotypes may meet the criteria for clinical obesity, highlighting the need for active medical intervention rather than passive lifestyle counseling alone.

## 2. Material and Methods

### 2.1. Study Population

This research was based on data gathered from the Korean National Health and Nutrition Examination Survey (KNHANES), which was carried out by the Korean Ministry of Health and Welfare from 2019 to 2021 [10]. Participants were selected through a stratified, multistage cluster sampling method, utilizing household registry information segmented by region, gender, and age category [11]. This nationally representative cross-sectional survey targeted non-institutionalized individuals and comprised detailed health interviews, nutritional evaluations, and physical examinations conducted by qualified personnel. The Institutional Review Board of Chosun University Hospital approved the study protocol (approval number: 2024-12-008, approved on 8 January 2025), and informed consent was provided by all participants.

A total of 22,948 individuals were initially identified, and after restricting the study population to adults aged ≥ 20 years (n = 17,780), we proceeded with the analysis. Participants with underlying chronic conditions were sequentially excluded, including those with chronic kidney disease (n = 211), hypertension (n = 3706), dyslipidemia (n = 1584), diabetes mellitus (n = 1183), thyroid disease (n = 753), asthma (n = 263), tuberculosis (n = 261), angina pectoris (n = 71), stroke (n = 70), myocardial infarction (n = 47), and liver cirrhosis (n = 17).

Moreover, individuals with missing data (n = 1131), those who had fasted less than 8 h prior to testing (n = 213), and those who were pregnant at the time of the survey (n = 43) were excluded. Subsequently, 8227 individuals without clinically diagnosed chronic diseases were included in the final analysis set. After excluding individuals with clinically diagnosed chronic diseases, we conducted our analysis on an apparently healthy adult population.

### 2.2. Measurement and Classification of Variables

Anthropometric measurements included height and weight assessed by a portable stadiometer and calibrated balance scale, respectively. Body mass index (BMI) was calculated as weight in kilograms divided by height in meters squared. Waist circumference was measured at the midpoint between the lowest rib margin and the iliac crest by trained examiners. Blood pressure (BP) measurements were obtained using a mercury sphygmomanometer after a 5 min seated rest; the mean value of two separate readings was used.

Following an overnight fast of at least 8 h, blood samples were obtained for analysis. Plasma glucose, high-density lipoprotein cholesterol (HDL-C), and triglyceride levels were measured using a Hitachi Automatic Analyzer 7600 (Hitachi High-Technologies Corporation, Tokyo, Japan), while glycated hemoglobin (HbA1c) was determined by high-performance liquid chromatography Tosoh G8 (Tosoh Corporation, Tokyo, Japan). Additionally, participants self-reported data on residential area, educational attainment, household income, smoking habits, alcohol consumption, physical activity, and total dietary energy intake.

Metabolic syndrome was identified when an individual met at least three of the following five conditions: (1) abdominal obesity, defined by a waist circumference ≥ 90 cm in men or ≥85 cm in women (as per the Korean Society for the Study of Obesity criteria); (2) elevated triglyceride levels (≥150 mg/dL) or treatment for hypertriglyceridemia; (3) reduced HDL-C levels (<40 mg/dL in men, <50 mg/dL in women, or ongoing lipid-lowering therapy); (4) elevated blood pressure (SBP ≥ 130 mmHg in men, DBP ≥ 85 mmHg in women, or use of antihypertensive drugs); and (5) impaired fasting glucose (≥100 mg/dL) or use of glucose-lowering medications.

Based on BMI and the presence or absence of metabolic syndrome, participants were stratified into four phenotype categories: MHNW (metabolically healthy normal weight): normal BMI (18.5–24.9 kg/m^2^) without metabolic syndrome; MANW (metabolically abnormal normal weight): normal BMI with metabolic syndrome; MHO (metabolically healthy obese): BMI ≥ 25 kg/m^2^ without metabolic syndrome; MAO (metabolically abnormal obese): BMI ≥ 25 kg/m^2^ with metabolic syndrome.

### 2.3. Definition of CKD

CKD was defined as either an estimated glomerular filtration rate (eGFR) < 60 mL/min/1.73 m^2^ or a urine albumin-to-creatinine ratio (UACR) > 30 mg/g based on the 2012 Kidney Disease: Improving Global Outcomes guidelines [12]. eGFR was calculated using the CKD-EPI equation.

### 2.4. Statistical Analysis

Statistical analyses were performed in accordance with the complex sampling design of the KNHANES, as recommended by the Korea Centers for Disease Control and Prevention. Continuous variables are presented as means with standard deviations (SD), while categorical variables are shown as weighted percentages. Group differences in general characteristics were assessed using chi-square tests. To explore factors linked to chronic kidney disease, multivariable logistic regression analyses were performed, adjusting for age, general and abdominal obesity, undiagnosed hypertension, undiagnosed dyslipidemia, family history of diabetes, sociodemographic characteristics (such as residential area, household income, and educational level), and lifestyle behaviors (including smoking status, alcohol use, physical activity, and overall caloric intake). To account for multiple comparisons across the six pairwise phenotype comparisons, Bonferroni correction was applied, setting the adjusted significance threshold at *p* < 0.0083. All statistical analyses were carried out using SPSS software version 25.0 (IBM Corp., Armonk, NY, USA). To evaluate potential multicollinearity among covariates included in the multivariable logistic regression models, we calculated variance inflation factors (VIFs). All VIF values were below 3.0, indicating no significant multicollinearity among variables (Appendix A).

## 3. Results

### 3.1. Basic Demographic and Clinical Characteristics

Baseline characteristics according to CKD status are presented in Table 1. A total of 8227 apparently healthy Korean adults were included in the final analysis. In the study population, the overall prevalence of chronic kidney disease (CKD) was 4.4% (n = 364); 95.6% (n = 7863) were classified as non-CKD. The CKD group was older and exhibited more adverse cardiometabolic profiles than the non-CKD group. The former had higher values for body mass index (BMI), waist circumference, blood pressure, fasting glucose, HbA1c, triglycerides, and markers of renal dysfunction. In contrast, the levels of high-density lipoprotein cholesterol (HDL-C), protein, fat, and total energy intake were lower in the CKD group. Notably, the proportion of metabolically abnormal phenotypes—specifically, MANW and MAO—was disproportionately higher in the CKD group compared to in the overall population. While MANW and MAO accounted for 5.0% and 13.2% of the total population, they represented 12.6% and 26.4% of the CKD group, respectively.

### 3.2. Clinical Characteristics and Chronic Kidney Status According to Body Size Phenotype

CKD prevalence according to body size phenotypes is shown in Table 2. The prevalence of CKD was 3.2% in the MHNW group, 10.5% in the MANW group, 4.0% in the MHO group, and 8.5% in the MAO group. A greater proportion of males had metabolic abnormalities or obesity. Individuals in the MANW group were older than those in other phenotypes. Kidney function, as measured by eGFR, was lower in the MAO and MANW groups. Additionally, members of the MANW group tended to have lower socioeconomic status and lower education levels and were more likely to reside in rural areas. The MANW phenotype had higher carbohydrate intake but lower total energy intake than the other phenotypes.

### 3.3. Association Between Chronic Kidney Status and Body Size Phenotype

In multivariable logistic regression analysis adjusted for age, sex, socioeconomic factors, lifestyle behaviors, and dietary intake, all non-MHNW phenotypes were significantly associated with increased risk of CKD compared to the MHNW reference group after Bonferroni correction for multiple comparisons (significance threshold: *p* < 0.0083) (Table 3).

The highest risk was observed in the MAO group, followed by the MANW and MHO groups. Specifically, MAO was associated with nearly a four-fold increase in CKD risk, and MANW with more than a two-fold increase. In pairwise comparisons, MAO had a higher risk of CKD than MHO, and the MANW group also showed a trend toward increased risk compared to MHO, although it did not reach statistical significance.

## 4. Discussion

In this nationally representative study of apparently healthy Korean adults, we found that body size phenotypes, defined by metabolic health status and body mass index (BMI), were significantly associated with the prevalence of CKD. In particular, the MAO and MANW phenotypes showed the highest prevalence of CKD. After adjusting for sociodemographic, lifestyle, and dietary factors, all non-MHNW phenotypes—MAO, MANW, and MHO—were associated with an increased risk of CKD compared to the MHNW reference group. Furthermore, the MAO phenotype was associated with a higher risk of CKD than the MHO phenotype. These findings suggest that both excess body weight and poor metabolic health contribute to the risk of CKD, with metabolic health playing a particularly critical role in determining renal risk.

Several previous studies have reported that metabolically unhealthy phenotypes, such as MAO and MANW, are significantly associated with an increased risk of CKD, consistent with our findings [13,14,15]. These results support the notion that metabolic health plays a more critical role than body size alone in determining CKD risk. However, in contrast to earlier studies, our study also found that the MHO phenotype was associated with an increased risk of CKD. This finding stands in contrast to previous reports that have regarded MHO as a relatively “benign” condition and highlights the need to reconsider the clinical implications of this phenotype.

Moreover, our study differs from previous research in several important aspects. While most prior studies were limited to patients with specific conditions such as diabetes [15] or were conducted in small city- [14] or hospital-based clinical populations [13], our study analyzed a nationally representative sample of apparently healthy Korean adults. This approach minimized potential confounding by known chronic diseases and enhanced the generalizability of our findings. We further reduced potential bias by excluding participants with previously diagnosed chronic diseases. In addition, our study defined CKD using both eGFR and UACR, which allowed us to detect early-stage renal impairment. Lastly, unlike some earlier studies that focused on the progression to end-stage kidney disease, our study highlights the importance of early identification and prevention strategies based on metabolic phenotyping.

Additionally, individuals in the MANW group, despite having normal body weight, were more likely to have lower socioeconomic status and lower educational attainment and reside in rural areas. They also exhibited imbalanced dietary patterns, characterized by high carbohydrate intake but low protein and fat consumption. These socioeconomic and nutritional disadvantages may contribute to underlying metabolic disturbances, thereby increasing the risk of CKD in this phenotype [16,17]. Importantly, metabolically unhealthy normal-weight individuals are often under-recognized due to their normal BMI, and economic barriers may limit their awareness of metabolic abnormalities or access to preventive healthcare [7,14]. These findings underscore that socioeconomic position—encompassing income, education, geographic location, and food environment—is a critical determinant of both metabolic and renal health. Individuals from low-income or rural backgrounds frequently face limited health literacy, reduced healthcare accessibility, and poor dietary quality, which can delay the detection and management of metabolic risk factors [18]. To effectively reduce CKD burden in these vulnerable populations, targeted public health interventions—such as community-based screening, nutritional education, and expanded access to preventive services—are essential. Collectively, our results highlight the need for a more comprehensive and socially informed approach to CKD prevention that extends beyond traditional clinical risk assessments.

Additionally, individuals in the MANW group, despite having normal body weight, were more likely to have lower socioeconomic status and lower educational attainment and reside in rural areas. They also exhibited imbalanced dietary patterns, characterized by high carbohydrate intake but low protein and fat consumption. These factors may contribute to metabolic disturbances and further increase the risk of CKD in this phenotype. These findings are consistent with recent studies suggesting that metabolically unhealthy normal-weight individuals represent an overlooked high-risk group [7,14]. However, because these individuals are not obese and often face economic constraints, they may not be aware of their underlying metabolic abnormalities. This highlights the need for targeted intervention strategies that take these contextual factors into account.

Furthermore, although the MHO phenotype has traditionally been regarded as relatively “benign”, recent studies have reported that this group is also associated with an increased risk of CKD. This elevated risk appears to be related to underlying biological mechanisms, particularly alterations in adipokines, rather than conventional metabolic markers alone. Specifically, hyperleptinemia has been shown to promote glomerular hyperfiltration, mesangial proliferation, and renal fibrosis, while hypoadiponectinemia contributes to inflammation, oxidative stress, and tubular injury—key processes in the pathophysiology of CKD [19,20,21]. A growing body of both preclinical and clinical evidence supports the role of adipokine dysregulation as a central mechanism in obesity-related kidney disease. Previous clinical studies and meta-analyses have also shown that individuals with the MHO phenotype have an increased risk of cardiovascular disease [22,23,24]. These findings suggest that even individuals with the MHO phenotype may not be metabolically or renally “healthy”, highlighting the need for active monitoring and intervention strategies that account for kidney risk in this group.

Although this study clearly demonstrated an association between adverse metabolic phenotypes and chronic kidney disease (CKD), the cross-sectional design limits the ability to determine the directionality of this relationship. On one hand, metabolic disturbances such as insulin resistance, dyslipidemia, and adipokine alterations may contribute to renal impairment [25,26]. On the other hand, emerging evidence suggests that early-stage CKD can, itself, induce systemic metabolic alterations through chronic inflammation, oxidative stress, and hormonal imbalances [27]. This potential bilateral relationship underscores the complex interplay between metabolic health and kidney function and highlights the need for prospective studies to elucidate the temporal and causal pathways linking these two conditions.

This study has several limitations that warrant acknowledgement. First, the cross-sectional design of the study precludes any inference of causality; thus, longitudinal studies are needed to establish the temporal and causal relationships between metabolic phenotypes and chronic kidney disease. Second, although we attempted to reduce potential confounding by excluding individuals with known underlying diseases, residual and unmeasured confounding—such as genetic predisposition, medication use, or variations in health awareness—may still exist. Third, certain variables—such as smoking status and physical activity—were derived from self-reported information, which may be prone to recall bias or inaccuracies in reporting. Fourth, metabolic health was defined using available clinical markers without more sophisticated assessments of insulin resistance or inflammatory status, which may limit the precision of phenotype classification. Last, excluding participants with self-reported chronic diseases may have led to an underestimation of CKD prevalence. Among the excluded individuals with available renal data, 10.1% were identified as having CKD, and they showed significantly higher rates of comorbid conditions such as hypertension and diabetes (Appendix A). However, the primary aim of this study was to identify early-stage CKD based on body size phenotypes in adults without known chronic conditions. This approach is meaningful in detecting high-risk individuals who may otherwise go unnoticed in primary care settings.

## 5. Conclusions

In this nationally representative study of healthy Korean adults, metabolically abnormal phenotypes—both obese (MAO) and normal weight (MANW)—were significantly associated with increased CKD risk. Notably, even the MHO group showed an elevated risk of CKD, challenging its previous “benign” classification. These results underscore the need to evaluate metabolic health and organ function rather than relying solely on body size when identifying individuals at high risk. A phenotype-based approach may improve early detection and guide targeted interventions for CKD prevention.

## Figures and Tables

**Table 1 biomedicines-13-01886-t001:** Baseline characteristics of the study population stratified by chronic kidney disease status.

	No CKD	CKD ^b^	*p*-Value
Number (%)	7863	364	
Sex (%)			
Male	3595 (45.7)	144 (39.6)	
Female	4268 (54.3)	220 (60.4)	0.024
Age (years)	43.98 ± 14.98	52.17 ± 16.60	0.001
BMI (kg/m^2^)	23.60 ± 3.68	24.56 ± 4.32	0.001
WC (cm)	81.77 ± 10.54	84.95 ± 11.59	0.001
SBP (mmHg)	115.08 ± 14.68	125.89 ± 19.84	0.001
DBP (mmHg)	74.82 ± 9.68	79.66 ± 12.80	0.001
FPG (mg/dL)	95.86 ± 14.43	107.23 ± 36.99	0.001
HbA1c (%)	5.55 ± 0.51	5.95 ± 1.24	0.001
LDL-C (mg/dL)	122.99 ± 33.48	120.17 ± 39.19	0.250
HDL-C (mg/dL)	53.90 ± 13.04	52.09 ± 12.61	0.005
TG (mg/dL)	123.48 ± 106.61	149.95 ± 174.61	0.001
Creatinine (mg/dL)	0.79 ± 0.16	0.81 ± 0.22	0.001
eGFR (mL/min/1.73 m^2^)	101.79 ± 18.06	96.29 ± 23.00	0.001
UACR (mg/g)	6.32 ± 4.29	100.88 ± 176.58	0.001
Family income percentile (%)			
<25	838 (10.7)	88 (24.3)	
25–50	1812 (23.2)	98 (27.1)	
50–75	2325 (29.7)	81 (22.4)	
≥75	2850 (36.4)	95 (26.2)	0.001
Education (%)			
More than high school education	6517 (87.3)	249 (75.7)	
Less than high school education	946 (12.7)	80 (24.3)	0.001
Residence (%)			
Urban area	6541 (83.2)	259 (71.2)	
Non-urban area	1322 (16.8)	105 (28.8)	0.001
Smoking			
Never	4699 (59.8)	228 (62.6)	
Past	1647 (20.9)	72 (19.8)	
Current	1478 (18.8)	62 (17.0)	0.728
Alcohol drinking			
Yes	4707 (59.9)	195 (53.6)	
No	3156 (40.1)	169 (46.4)	0.017
Regular exercise ^a^			
Yes	3481 (46.6)	147 (44.7)	
No	3982 (53.4)	182 (55.3)	0.484
Total energy intake (kcal)	1926.75 ± 866.82	1824 ± 826.92	0.001
Carbohydrate intake (g)	268.67 ± 110.78	269.95 ± 113.86	0.845
Protein intake (g)	73.14 ± 39.71	65.57 ±34.66	0.001
Fat intake (g)	51.8 ± 37.15	43.0 ± 33.36	0.001
Metabolic phenotype (%)			
MHNW	5093 (64.8)	166 (45.6)	
MANW	394 (5.0)	46 (12.6)	
MHO	1337 (17.0)	56 (15.4)	
MAO	1039 (13.2)	96 (26.4)	0.001

Data are expressed as the mean ± standard deviation (SD) for continuous variables and as weighted percentages for categorical variables. BMI, body mass index; DBP, diastolic blood pressure; eGFR, estimated glomerular filtration rate; FPG, fasting plasma glucose; MANW, metabolically abnormal normal weight; MAO, metabolically abnormal obese; MHNW, metabolically healthy normal weight; MHO, metabolically healthy obese; SBP, systolic blood pressure; TG, triglyceride; UACR, urine albumin-to-creatinine ratio; WC, waist circumference. ^a^ Regular exercise was defined as participating in a minimum of 2.5 h of moderate-intensity physical activity or 1.25 h of high-intensity activity each week. ^b^ defined as UACR > 30 mg/g and/or eGFR < 60 mL/min/1.73 m^2^ estimated by the CKD-EPI equation.

**Table 2 biomedicines-13-01886-t002:** Baseline characteristics of the study population by body size phenotype.

	MHNW	MANW	MHO	MAO	*p*-Value
Number (%)	5259 (63.9)	440 (5.3)	1393 (16.9)	1135 (13.7)	
Sex (%)					
Male	2044 (38.8)	238 (53.1)	811 (58.2)	718 (63.2)	0.001
Female	3349 (62.1)	210 (46.9)	606 (42.8)	427 (37.3)	0.001
Age (years)	43.45 ± 15.15	56.78 ± 13.54	41.98 ± 14.40	46.80 ± 14.00	0.001
BMI (kg/m^2^)	21.58 ± 2.03	23.13 ± 1.51	27.44 ± 2.44	28.67 ± 3.22	0.001
WC (cm)	76.24 ± 7.02	84.75 ± 6.32	91.00 ± 7.54	95.91 ± 7.53	0.010
SBP (mmHg)	111.64 ± 13.76	129.74 ± 16.74	117.00 ± 12.00	126.38 ± 14.99	0.001
DBP (mmHg)	72.57 ± 8.98	81.02 ± 10.23	75.94 ± 8.46	82.87 ± 10.17	0.001
FPG (mg/dL)	93.00 ± 11.26	110.14 ± 28.19	94.84 ± 10.17	108.48 ± 24.92	0.001
HbA1c (%)	5.46 ± 0.39	5.98 ± 1.00	5.52 ± 0.36	5.93 ± 0.90	0.001
LDL-C (mg/dL)	120.68 ± 34.30	118.64 ± 35.46	132.06 ± 30.50	123.85 ± 33.16	0.523
HDL-C (mg/dL)	57.39 ± 12.70	43.50 ± 10.01	52.32 ± 10.58	43.30 ± 9.72	0.001
TG (mg/dL)	97.31 ± 64.79	236.91 ± 217.76	113.92 ± 70.93	220.03 ± 162.78	0.001
Creatinine (mg/dL)	0.77 ± 0.16	0.80 ± 0.18	0.82 ± 0.16	0.83 ± 0.16	0.002
eGFR (mL/min/1.73 m^2^) ^a^	102.58 ± 18.12	96.74 ± 19.32	101.15 ± 17.71	99.33 ± 19.07	0.414
UACR (mg/g)	9.47 ± 43.27	17.7 ± 70.52	8.47 ± 17.70	14.95 ± 42.53	0.001
Family income percentile (%)					
<25	568 (10.8)	79 (21.8)	135 (9.7)	136 (12.0)	
25–50	1146 (21.8)	128 (29.2)	358 (25.7)	291 (25.7)	
50–75	1557 (29.6)	93 (21.1)	430 (30.9)	339 (29.9)	
≥75	1988 (37.8)	120 (27.9)	460 (33.7)	369 (32.5)	0.001
Education (%)					
More than high school education	4537 (86.2)	274 (62.2)	1203 (86.3)	894 (78.7)	
Less than high school education	856 (16.1)	174 (39.5)	214 (15.3)	251 (22.1)	0.001
Residence (%)					
Urban area	4433 (84.3)	303 (69.0)	1154 (82.9)	904 (79.7)	
Non-urban area	826 (15.7)	137 (31.0)	239 (17.1)	231 (20.3)	0.001
Smoking					
Never	3413 (64.9)	216 (49.3)	744 (53.4)	549 (48.4)	
Past	962 (18.3)	112 (25.4)	350 (25.1)	296 (26.1)	
Current	884 (16.2)	112 (24.6)	299 (21.2)	290 (25.1)	0.001
Alcohol drinking					
No	2183 (41.5)	192 (43.5)	513 (36.8)	445 (39.2)	
Yes	3076 (58.5)	248 (56.5)	880 (63.2)	690 (60.8)	0.006
Regular exercise ^a^					
No	2.803 (53.3)	284 (64.6)	659 (47.3)	648 (57.1)	
Yes	2456 (46.7)	156 (35.4)	734 (52.7)	487 (42.9)	0.001
Total energy intake (kcal)	1877.20 ± 822.24	1955.24 ± 848.89	1996.67 ± 886.45	2029.18 ± 1004.35	0.001
Carbohydrate intake (g)	264.78 ± 108.06	289.38 ± 109.44	270.64 ± 115.33	278.55 ± 118.32	0.001
Protein intake (g)	70.92 ± 37.68	70.06 ± 37.68	77.52 ± 41.09	76.73 ± 45.29	0.001
Fat intake (g)	51.06 ± 36.21	43.50 ± 33.07	54.54 ± 37.66	52.20 ± 40.97	0.001
CKD diagnosis ^b^	168 (3.2)	46 (10.5)	56 (4.0)	96 (8.5)	0.001

Data are expressed as the mean ± standard deviation (SD) for continuous variables and as weighted percentages for categorical variables. BMI, body mass index; DBP, diastolic blood pressure; eGFR, estimated glomerular filtration rate; FPG, fasting plasma glucose; MANW, metabolically abnormal normal weight; MAO, metabolically abnormal obese; MHNW, metabolically healthy normal weight; MHO, metabolically healthy obese; SBP, systolic blood pressure; TG, triglyceride; UACR, urine albumin-to-creatinine ratio; WC, waist circumference. ^a^ Regular exercise was defined as participating in a minimum of 2.5 h of moderate-intensity physical activity or 1.25 h of high-intensity activity each week. ^b^ defined as UACR > 30 mg/g and/or eGFR < 60 mL/min/1.73 m^2^ estimated by the CKD-EPI equation.

**Table 3 biomedicines-13-01886-t003:** Association between chronic kidney disease and body size phenotypes.

			Fully Adjusted OR (95% Cl)					
	MAO	*p*-Value *		MANW	*p*-Value *		MHO	*p*-Value *
MAO/MHNW			MANW/MHNW			MHO/MHNW		
CKD (−)	Reference		CKD (−)	Reference		CKD (−)	Reference	
CKD (+)	3.770 (2.648–5.367)	<0.001	CKD (+)	2.492 (1.547–4.016)	<0.001	CKD (+)	1.974 (1.358–2.870)	0.005
MAO/MANW			MANW/MHO					
CKD (−)	Reference		CKD (−)	Reference				
CKD (+)	1.242 (0.757–2.037)	0.539	CKD (+)	1.477 (0.977–2.232)	0.124			
MAO/MHO								
CKD (−)	Reference							
CKD (+)	1.897 (1.221–2.945)	0.014						

CI, confidence interval; MANW, metabolically abnormal normal weight; MAO, metabolically abnormal obese; MHNW, metabolically healthy normal weight; MHO, metabolically healthy obese; OR, odds ratio. The logistic regression models were adjusted for age, sex, sociodemographic variables (residential area, household income, and education level), and lifestyle factors (smoking status; alcohol use; regular physical activity; total energy intake; and macronutrient intake, including carbohydrates, protein, and fat); * Statistical significance was set at *p* < 0.0083 using Bonferroni correction for multiple comparisons.

## Data Availability

Data Availability Statement: The data utilized in this study are publicly accessible through the Korea National Health and Nutrition Examination Survey (KNHANES), managed by the Korea Centers for Disease Control and Prevention (KCDCP). They can be freely retrieved from the KCDCP website (https://knhanes.cdc.go.kr), accessed on 2 June 2025.

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
