# Peer review of "Association Between Incident Chronic Kidney Disease and Body Size Phenotypes in Apparently Healthy Adults: An Observational Study Using the Korean National Health and Nutrition Examination Survey (2019–2021)"

_biomedicines, 2025, doi:10.3390/biomedicines13081886_

Round 1

Reviewer 1 Report

Comments and Suggestions for Authors

This is an interesting article drawing the relation between the occurence of chronic kidney disease, body size phenotypes and metabolic health.

I have several commments: 

1. In lines 89-94, I read that the data set started from 22948 individuals, was reduced to 17780 by restricting for age, and then reduced to 8227 individuals after exclusion for chronic diseases. 

There are two things that I do not understand and that are important to avoid research bias :  more than half of those 17780 patients were excluded because they had a chronic disease.  
2. Can the authors confirm that more than half of the Korean population from 20 years or older that participated in the study have a chronic disease ?
3. Secondly, CKD is a chronic disease, but patients suffering from chronic diseases, even those with renal failure (that may have been caused by CKD) were excluded.  Can the authors give an estimate of the proportion of patients with CKD that were excluded, for example when they had one or more of other chronic diseases ?  
Certainly because this manuscript looks at factors like obesity and metabolic health, which are themselves related to chronic diseases, it is important to assess the bias that may have been cause by leaving out more than half of the patients.

4. In line 128, I see periodontitis mentioned.  What does periodontitis have to do with CKD ?

5. Table 1 shows, for the discontinuous variables, the percentage of CKD and not CKD cases for every category, but not for the variables that consist of two catogeries. I know that these percentages can be calculated from the numbers that are given, but could the authors also add the distribution CKD/not CKD for femails, for peole that did not finish high school, for people that don't live in an urban area, for those that do not drink alcohol or regularly excercise ?

6. Was the significance level in the confidence intervals shown in Table 3 corrected for simultaneous hypothesis testing ?

7. If I counted well, I see that a total of 13 covariates were included in the multivariate logistic regression model, among which physical activity, total dietary energy intake and abdominal obesity.  I expect that those factors are highly correlated with body size phenotype, possibly multicollinearity in the model. For your information, multicollinearity occurs when highly correlated variables are put together in a (logistic) regression model and can be identified by variance inflation factors.  I don't see anything mentioned in the manuscript that this issue has been handled by the authors.  I strongly recommend to assess the multicollinearity of the model.

Author Response

Lyu et al., Association between incident chronic kidney disease and body size phenotypes in apparently healthy adults: An Observational Study Using the Korean National Health and Nutrition Examination Survey (2019–2021)

<Response to reviewers and summary of changes made>

             We thank the reviewers for their careful review of our manuscript. We found their comments and suggestions very helpful. We have tried to address all of the reviewers’ comments in our revised manuscript. Amended and newly added content is shown in red font.

Please refer to the Word file

Reviewer 2 Report

Comments and Suggestions for Authors

This manuscript presents a cross-sectional analysis of 8,227 Korean adults from KNHANES 2019–2021, examining the association between chronic kidney disease (CKD) and body size phenotypes characterized by metabolic health and BMI.  The authors demonstrate that metabolically abnormal phenotypes—particularly metabolically abnormal obese (MAO) and metabolically abnormal normal weight (MANW)—exhibit a significant correlation with increased probabilities of chronic kidney disease (CKD), even among persons devoid of overt chronic illnesses.  Metabolically healthy obesity (MHO) significantly elevates the risk of chronic kidney disease (CKD), hence questioning its purportedly "benign" characteristics.

 The subject is pertinent and current, and the work is predominantly well-crafted and methodologically robust.  Nonetheless, certain sections necessitate elucidation, supplementary context, and corrections.

The authors appropriately recognize the cross-sectional design as a drawback. Nevertheless, the Discussion might more explicitly address the inability to ascertain directionality—specifically, whether metabolic abnormalities lead to chronic kidney disease (CKD) or whether early CKD induces metabolic alterations.

This study defines metabolic health primarily in terms of glycemic markers, specifically fasting plasma glucose (FPG) and hemoglobin A1c (HbA1c). This is more limited than definitions that encompass blood pressure, lipids, and inflammatory markers commonly utilized in metabolic phenotyping. The authors should have:

Provide a rationale for the exclusive utilization of glycemic parameters.

Examine the possible misclassification or underestimate of metabolic unhealthiness.

The manuscript delineates notable sociodemographic disparities (income, education, urban/rural residency) among phenotypes. Incorporating the following discussions would enhance the paper:

Potential connections between socioeconomic position, metabolic health, and chronic kidney disease (CKD).

How interventions may vary in low-income or rural populations.

Abstract:

“multivariable logistic regression models adjusted for age, sex, sociodemographic factors, lifestyle behaviors, and diet was used.”  Should be “were used.”

Introduction:

Lines 43–46: The sentence on the definition of obesity from the Lancet Commission is quite dense. Consider simplifying for clarity.

Methods:

Logistic regression analyses were performed to identify characteristics linked with periodontitis (page 3, line 127). This seems to be a remnant from a different manuscript. It should be amended to "associated with CKD."

Results:

Some tables include means ± SD while others include SE. This confusion must be corrected for the presentation of findings.

Typographical Errors:

Several minor typos (e.g., spacing issues) appear throughout, but these are minor and easily fixed during copyediting.

Author Response

(The authors gave the same response as above.)

Round 2

Reviewer 1 Report

Comments and Suggestions for Authors

I thank the authors for modifying the manuscript. 

I have one supplementary question. I understand now that the study was performed on apparently healthy individuals. Nevertheless, I do not see that explicitly mentionned in the manuscript - at least not in the sectio where the authors explain which data were left out. Can the authors still add a sentence near the section where they explain the data rhat were left out, that the target population of this study are apparently healthy individuals ?

Author Response

Lyu et al., Association between incident chronic kidney disease and body size phenotypes in apparently healthy adults: An Observational Study Using the Korean National Health and Nutrition Examination Survey (2019–2021) _ Secondary revision

<Response to reviewers and summary of changes made>

             We thank the reviewers for their careful review of our manuscript. We found their comments and suggestions very helpful. We have tried to address all of the reviewers’ comments in our revised manuscript. Amended and newly added content is shown in red font.

Reviewer #1

Comment 1) I have one supplementary question. I understand now that the study was performed on apparently healthy individuals. Nevertheless, I do not see that explicitly mentionned in the manuscript - at least not in the sectio where the authors explain which data were left out. Can the authors still add a sentence near the section where they explain the data rhat were left out, that the target population of this study are apparently healthy individuals ?:

Thank you for your thoughtful comment and for pointing out the need to clarify the study population. As suggested, we have revised the Methods section to explicitly state that our target population consisted of apparently healthy individuals. Specifically, we added the following sentence to the paragraph describing the exclusion criteria:

Newly inserted description (Page 3 Line 99-101)

After excluding individuals with clinically diagnosed chronic diseases, we focused our analysis on an apparently healthy adult population.

Reviewer 2 Report

Comments and Suggestions for Authors

I am satisfied with the author’s responses to my questions/issues raised in my initial review. The revised manuscript is easier to follow based on feedback from the reviewers. I recommend that the revised manuscript be accepted.

Author Response

Lyu et al., Association between incident chronic kidney disease and body size phenotypes in apparently healthy adults: An Observational Study Using the Korean National Health and Nutrition Examination Survey (2019–2021)

<Response to reviewers and summary of changes made>

             We thank the reviewers for their careful review of our manuscript. We found their comments and suggestions very helpful. We have tried to address all of the reviewers’ comments in our revised manuscript. Amended and newly added content is shown in red font.

Reviewer #2

I am satisfied with the author’s responses to my questions/issues raised in my initial review. The revised manuscript is easier to follow based on feedback from the reviewers. I recommend that the revised manuscript be accepted.

Reply) We sincerely appreciate your positive evaluation of our revision. We are grateful for your helpful feedback during the review process, which has significantly improved the clarity and quality of our manuscript. Thank you for recommending our revised manuscript for acceptance.
